# A Multifunctional Hybrid Nanocarrier for Non-Invasive siRNA Delivery to the Retina

**DOI:** 10.3390/pharmaceutics15020611

**Published:** 2023-02-11

**Authors:** Shogo Nishida, Yuuki Takashima, Ryotaro Udagawa, Hisako Ibaraki, Yasuo Seta, Hiroshi Ishihara

**Affiliations:** School of Pharmacy, Tokyo University of Pharmacy and Life Sciences, 1432-1 Horinouchi, Hachioji, Tokyo 192-0392, Japan

**Keywords:** siRNA, functional peptide, retina pigment epithelium cells, siRNA delivery, instillation, VEGF

## Abstract

Drug therapy for retinal diseases (e.g., age-related macular degeneration, the leading cause of blindness) is generally performed by invasive intravitreal injection because of poor drug delivery caused by the blood–retinal barrier (BRB). This study aimed to develop a nanocarrier for the non-invasive delivery of small interfering RNA (siRNA) to the posterior segment of the eye (i.e., the retina) by eyedrops. To this end, we prepared a hybrid nanocarrier based on a multifunctional peptide and liposomes, and the composition was optimized. A cytoplasm-responsive stearylated peptide (STR-CH2R4H2C) was used as the multifunctional peptide because of its superior ability to enhance the complexation, cell permeation, and intracellular dynamics of siRNA. By adding STR-CH2R4H2C to the surface of liposomes, intracellular uptake increased regardless of the liposome surface charge. The STR-CH2R4H2C-modified cationic nanocarrier demonstrated significant siRNA transfection efficiency with no cytotoxicity, enhanced siRNA release from endosomes, and effectively suppressed vascular endothelial growth factor expression in rat retinal pigment epithelium cells. The 2.0 mol% STR-CH2R4H2C-modified cationic nanocarrier enhanced intraocular migration into the retina after instillation into rat eyes.

## 1. Introduction

Retinal diseases, such as age-related macular degeneration and diabetic retinopathy, are the leading causes of blindness and failing vision. The number of patients with retinal disease increases with aging and lifestyle-related diseases [1,2,3,4]. The retina is damaged by abnormal angiogenesis, which causes degeneration in the choroidal–retinal pigment epithelial in early disease stages [2,5,6,7]. Inhibitors of vascular endothelial growth factor (VEGF), which plays a role in neovascularization, are used to treat retinal diseases. Retinal pigment epithelium (RPE) cells are located in the outermost layers of the retina and form tight junctions, which function as an outer blood–retinal barrier (BRB) to prevent the entry of foreign substances from the outside; however, the BRB makes it difficult to treat retinal diseases by systemic circulation and eyedrops [7,8]. Hence, most treatment for retinal disease is by invasive intravitreal injection with the risk of physical and mental burden, infection, and elevated intraocular pressure [2,9,10,11]. Therefore, it is important to develop drug delivery systems that allow non-invasive instillation.

Choroidal capillaries have a fenestrated structure with a pore size of around 70–80 nm to provide nutrition to the inside of the retina [8,12,13,14]. We previously demonstrated that particle size contributed to delivery efficiency, and liposomes with a size of 80 nm or less improved intraocular distribution to the posterior segment of the eye after instillation in rats. In addition, we showed that a transferrin modification to liposomes, which targets transferrin receptors expressed on RPE cells, improved accumulation in the rat eye after instillation [8,15]. Generally, a drug instilled into the eye is quickly drained through the nasolacrimal duct with tear fluid. For effective small interfering RNA (siRNA) delivery, it was important to construct a carrier that could improve migration in the target tissue and control in vivo and intracellular dynamics. Carriers, such as 1,2-dioleoyl-sn-glycero-3-phosphocholine (DOPC)/cholesterol (CHO)-based cationic liposomes and anionic 1,2-dioleoyl-sn-glycero-3-phosphatidylethanolamine (DOPE)/cholesteryl hemisuccinate (CHEMS) liposomes, enhance the permeation of siRNA in the skin. Cationic liposomes show higher penetration with an increase in positive surface charge; however, the high positive charge causes skin irritation and cytotoxicity [16].

Cell-penetrating peptide (CPP) enhances intracellular transduction and tissue permeability of low- and high-molecular-weight compounds [17,18,19,20]. Polymer micelles composed of a typical CPP (e.g., HIV-Tat-derived peptide) and amphiphilic block copolymers enhance migration to the brain through the nasal mucous membrane [21,22]. Various CPPs based on oligo-arginine and oligo-lysine have high intracellular penetration through a strong interaction with the cell surface and form stable complexes due to the electrostatic interactions between the cationic CPP and the anionic phosphate group of nucleic acids [23,24,25,26,27,28]. We previously demonstrated that cytoplasm-responsive stearylated peptide STR-CH2R4H2C is a multifunctional peptide that enhances the cell permeation and intracellular dynamics of siRNA. STR-CH2R4H2C consists of arginine (R), histidine (H), cysteine (C), and stearic acid (STR) [29,30,31]. Arginine promotes high cellular uptake. Histidine allows the escape from the endosome due to a proton sponge effect, and cysteine and arginine ensure stable complex formation with siRNA and its release in the cytoplasm [29,32,33,34]. It has been reported that STR-CH2R4H2C improves the intracellular dynamics of siRNA and inhibits NF-*k*B activity related to inflammatory cytokine production in mouse models of rheumatoid arthritis and atopic dermatitis [32,33,35].

This study aimed to identify a nanocarrier that could enhance intracellular dynamics and intraocular migration following instillation into the eye. To this end, we prepared a hybrid nanocarrier consisting of the multifunctional STR-CH2R4H2C peptide and liposomes. The intracellular uptake, siRNA transfection, and silencing effects of this hybrid carrier were evaluated in RPE cells, and its intraocular behavior was investigated in rat eyes following instillation.

## 2. Materials and Methods

### 2.1. Reagents and Cells

The STR-CH2R4H2C peptide (CH3(CH2)_16_CO-Cys-His-His-Arg-Arg-Arg-Arg-His-His-Cys-OH, MW 1664.1) was specially ordered from BEX Co., Ltd. (Tokyo, Japan) [29,30]. DOPC (#850375P), 1,2-dioleoyl-3-trimethylammonium-propane (DOTAP; #890890C) and CHO (#700100P) were purchased from Avanti Polar Lipids (Alabaster, AL, USA). LipoTrust^TM^ *EX* Oligo (LipoTrust; #LEO-01) was purchased from Hokkaido System Science Co., Ltd. (Hokkaido, Japan). ATTO647N DOPE (ATTO-DOPE; #AD647-16), used in the fluorescent labeling of liposomes, was purchased from ATTO-TEC (Siegen, Germany). The 6-carboxyfluorescein-aminohexylphosphoramidite (FAM)-labeled siRNA (FAM-siRNA; sense: 5′-6-carboxyfluorescein AUC CGC GCG AUA GUA CGU AdTdT-3′), Cyanine 5-labeled siRNA (Cy5-siRNA; sense, 5′-Cyanin5 AUC CGC GCG AUA GUA CGU AdTdT-3′), and rat VEGF siRNA (siVEGF; sense: 5’-CUU CCA GAA ACA CGA CAA AdTdT-3′) were obtained from Bioneer Co. (Daejeon, Korea). Negative control siRNA (siControl; BIN SN-1003, AccuTarget^TM^ Negative Control siRNA) was purchased from Thermo Fisher Scientific (Waltham, MA, USA). Special-grade chloroform, isopropanol, and ethanol were from Nacalai Tesque (Kyoto, Japan).

Rat retinal pigment epithelial cells (RPE-J, CRL-2240^TM^) were obtained from the American Type Culture Collection (Manassas, VA, USA). Dulbecco’s Modified Eagle Medium (DMEM; #08458-16, high glucose, Nacalai Tesque, Kyoto, Japan), fetal bovine serum (FBS; #SH30910.03, Cytiva, Tokyo, Japan), penicillin (100 U/mL)-streptomycin (100 µg/mL) solution (#2625384, Nacalai Tesque, Kyoto, Japan), 1% non-essential amino acids solution (NEAA, #0634456, Nacalai Tesque, Kyoto, Japan), heparin (FUJIFILM Wako Pure Chemical, Osaka, Japan), LysoTracker^TM^ Red DND-99 (#L7528, Thermo Fisher Scientific, Waltham, MA, USA), SlowFade^®^ antifade reagent (#S36937, Thermo Fisher Scientific, Waltham, MA, USA), Hoechst^®^ 33342 (#H342, DOJINDO Laboratories, Kumamoto, Japan), and Opti-MEM^TM^ (#31985070, Thermo Fisher Scientific, Waltham, MA, USA) were used in the cytological assays. The DC^TM^ Protein Assay Kit was purchased from Bio-Rad (#5000112JA, CA, USA). The rat VEGF enzyme-linked immunoassay (ELISA) Kit was purchased from R&D Systems (#RRV00, Minneapolis, MN, USA). TRIzol^TM^ (#12183555, Thermo Fisher Scientific Waltham, MA, USA), Prime Script^TM^ RT reagent Kit (#RP037B, Takara Bio, Shiga, Japan), GoTaq^®^ qPCR Master Mix (#A6001, Promega, WI, USA), TaqMan^TM^ Gene Expression Assay primers for rat glyceraldehyde 3-phosphate dehydrogenase (GAPDH), and rat VEGF (#4331182, Thermo Fisher Scientific, Waltham, MA, USA) were used for reverse transcription-quantitative polymerase chain reaction (RT-qPCR). Dako fluorescence mounting medium was purchased from Agilent Technologies, Ltd. (#S3023, Santa Clara, CA, USA).

### 2.2. Preparation of STR-CH2R4H2C-LP and STR-CH2R4H2C-LP-siRNA Lipoplexes

Figure 1 shows the preparation procedure for STR-CH2R4H2C-modified liposomes (STR-CH2R4H2C-LP) and siRNA-loaded STR-CH2R4H2C-LP (STR-CH2R4H2C-LP-siRNA lipoplexes). All lipids were dissolved in chloroform (5 mg/mL) as stock solutions and stored at −20 °C until use. LP were prepared using the thin film method. Briefly, lipid solutions were mixed at different molar ratios (Table 1) and then dried with a rotary evaporator to form a thin lipid film. The thin films were hydrated with 10 mM HEPES (pH 8.0) and then sonicated (BRANSON Digital Sonifier^®^; Central Scientific Commerce, Tokyo, Japan) for 3 min at 30 W, yielding liposomes with different surface charges [36]. STR-CH2R4H2C was dissolved in 10 mM HEPES buffer (pH 7.4) and gently dropped into the liposomes while vortexing (1.0 or 2.0 mol% STR-CH2R4H2C to total lipid). The unmodified free peptides were removed by ultrafiltration (13,000× *g* for 3 min, Amicon^®^ ultra 100K; Sigma-Aldrich, St. Louis, MO, USA) to obtain STR-CH2R4H2C-LP [37]. Subsequently, siRNA was added to STR-CH2R4H2C-LP at a molar ratio of nitrogen on lipid and peptide to phosphate on siRNA (N/P ratio) of 5:1. The STR-CH2R4H2C-LP-siRNA lipoplexes were obtained by removing free siRNAs by ultrafiltration (13,000× *g* for 1 min, Amicon^®^ ultra 100 K).

### 2.3. Physical Properties of the STR-CH2R4H2C-LP-siRNA Lipoplexes

STR-CH2R4H2C-LP-siRNA lipoplexes were diluted to an appropriate concentration. Mean particle size, polydispersity index (PDI), and zeta potential were measured using a Zetasizer Nano ZSP (Malvern Panalytical, UK). SYBR^®^ GREEN solution (50 µL) was added to the STR-CH2R4H2C-LP-siRNA lipoplexes (50 µL), and then the mixture was incubated for 15 min at room temperature. The fluorescence intensity of each sample was measured using a microplate reader (Ex:494 nm, Em:521 nm, Varioskan Flash 2.4, Thermo Fisher Scientific, MA, USA). The siRNA amount in the lipoplexes was determined using calibration curves of siRNA standard solutions (0.125–4 µg/100 µL). The siRNA loading efficiency (%) was calculated as the percentage of the measured amount to the formulated amount of siRNA.

### 2.4. Cellular Uptake of STR-CH2R4H2C-LP-siRNA Lipoplexes in RPE Cells

RPE cells (2.0 × 10^5^ cells/well) in 24-well plates were precultured in DMEM with 10% FBS and 1% NEAA for 24 h at 33 °C in a humidified 5% CO_2_ atmosphere. The cells were washed with phosphate-buffered saline (PBS) and treated with 100 µL STR-CH2R4H2C-LP-siRNA lipoplexes (FAM-siRNA or Cy5-siRNA 0.158 µM or ATTO-DOPE 0.2 µM) in 900 µL serum-free DMEM for 4 h at 33 °C at 5% CO_2_. The cells were washed in PBS containing 20 U/mL heparin to remove the lipoplexes attached to the surface of the cells. After treatment with 0.25% Trypsin-EDTA, the cells were collected in a test tube [38]. Fluorescence intensities were measured using flow cytometry (FACS Celesta; Becton Dickinson, NJ, USA). The cellular uptake of STR-CH2R4H2C-LP-siRNA lipoplexes and siRNA were evaluated by measuring the fluorescence intensity in 10,000 events per sample.

### 2.5. Evaluation of siRNA Intracellular Dynamics

RPE cells (3.0 × 10^5^ cells/well) were grown on cover glasses in 6-well plates for 24 h at 33 °C and 5% CO_2_. After washing cells in PBS, 100 µL STR-CH2R4H2C-LP-siRNA lipoplexes (Cy5-siRNA, 0.079 µM) was added to the cells in 1900 µL serum-free DMEM. This study examined samples in which siRNA transfection into cells was confirmed by flow cytometry. The objective of this study is to confirm internalization of the lipoplex into the cells which is treated with peptide-tacked lipoplex. The cells were incubated for 4 h at 33 °C and 5% CO_2_. The cells were washed with PBS containing heparin (20 U/mL). To stain the endosomes (lysosomes), the cells were incubated with serum-free DMEM containing 75 nM LysoTracker^TM^ Red DND-99 for 1 h [39]. After washing with PBS, the cells were fixed with 4% paraformaldehyde and sealed on glass slides with SlowFade^®^ containing 1% Hoechst^®^ 33342. The fluorescence of the endosomes and Cy5-siRNA was observed using a confocal laser scanning microscope (FV1000D IX81; OLYMPUS, Tokyo, Japan).

### 2.6. Suppression of VEGF Expression in RPE Cells with STR-CH2R4H2C-LP-siVEGF Lipoplexes

RPE cells (3.0 × 10^5^ cells/well) were grown in 6-well plates for 24 h at 33 °C and 5% CO_2_. After washing with PBS, 100 µL STR-CH2R4H2C-LP-siRNA lipoplexes (siVEGF or siControl, 0.079 µM) and LipoTrust (siVEGF or siControl, 0.079 µM) were added to the cells in 1900 µL Opti-MEM^TM^, and then the cells were incubated for 4 h at 33 °C and 5% CO_2_. After washing the cells with PBS containing 20 U/mL heparin, the cells were incubated with 2000 µL Opti-MEM^TM^ for 72 h at 33 °C and 5% CO_2_. These conditions enable the induction of VEGF production in normal RPE cells under low nutrient and oxidative stress [40]. The amounts of secreted VEGF in the medium and total protein of the cells were quantified using a rat VEGF ELISA Kit and DC^TM^ Protein assay, respectively. The amount of VEGF was normalized to the total protein content. Subsequently, the relative VEGF protein expression levels were calculated using the following equation: VEGF protein expression levels (%) = VEGF in treated cells × 100/VEGF in untreated cells (control).

In addition, VEGF mRNA expression levels were measured. After 72 h, treated cells were washed with PBS. To isolate RNA, cells were lysed with 1 mL TRIzol^TM^. Total RNA was isolated using TRIzol^TM^ reagent according to the manufacturer’s instructions, and then the RNA pellet was resuspended in nuclease-free water to a concentration of 100 ng/µL [41]. The cDNA was reverse transcribed using the Takara PCR Thermal Cycler Dice^TM^ (reaction cycle; 37 °C 15 min, 85 °C 5 min, 4 °C 5 min; Takara Bio, Shiga, Japan). Briefly, RNA (1000 ng) was mixed with 10 µL Prime Script^TM^ RT Reagent Kit solution (4 µL Prime Script Buffer, 1 µL oligo dT primer, 4 µL random hexamers). Real-time qPCR was performed using a PCR thermal cycler (LineGene; NIPPON Genetics Tokyo, Japan). The qPCR reaction mix consisted of 9 µL cDNA, 10 µL GoTaq^®^ qPCR Master Mix, and 1 µL of each GAPDH or VEGF primer. PCR amplification was carried out for 55 cycles at 95 °C for 10 s, 58 °C for 10 s, and 72 °C for 20 s. The amplification curve was analyzed, and the ΔCt values were determined (Ct_VEGF_ − Ct_GAPDH_). ΔΔCt was calculated as ΔCt_sample_ − ΔCt_control_. The relative VEGF mRNA expression level (%) was determined based on 2^−ΔΔCt^ × 100 [42].

### 2.7. Intraocular Migration of STR-CH2R4H2C-LP Following Instillation into the Rat Eye

Twelve-week-old male Sprague Dawley rats were purchased from SLC (Shizuoka, Japan). All animal experiments were conducted in accordance with protocols (#P21-39, #P22-64) approved by the Animal Care and Ethics Committee of the Tokyo University of Pharmacy and Life Sciences. The rats were housed under standard conditions (23.5 ± 1 °C, 55 ± 5% relative humidity, 12 h light/dark cycles). Food and water were supplied ad libitum.

To evaluate the intraocular migration of STR-CH2R4H2C-LP in the rat eye, 20 µL STR-CH2R4H2C-LP (Total lipid 0.60 µmol/20 µL) was dropped into the right eye of anesthetized rats. After 2 h, the eyeballs were removed and fixed in 4% paraformaldehyde for 3 h. The retina was isolated from the eyeball, rinsed with saline, and then cut into four sections for flat mounting on glass slides. The fluorescence of ATTO-labeled liposomes was observed using a fluorescence microscope (BZ8000; KEYENCE, Osaka, Japan). The exposure time was set using the untreated left eye of each rat as a negative control. For histological observation, 30 µL DOTAP0.250-LP or 2.0 mol% STR-CH2R4H2C-DOTAP0.250-LP (Total lipid 0.18 µmol/30 µL) was dropped into the right eye of anesthetized rats. After 2 h, the eyeballs were removed and fixed in Davidson fixing solution (composition ratio, 4% paraformaldehyde/99.5% ethanol/acetic acid = 5/3/1, *v*/*v*/*v*) for 4 h at 4 °C. Subsequently, the eyeballs were treated with sucrose solutions (10% soln. for 4 h; 15% soln. for 4 h; 20% soln. overnight) at 4 °C, and then freeze-embedded using tissue mount agent (TISSU MOUNT^®^, Chiba Medical, Saitama, Japan). Frozen tissue blocks were sectioned into a 10 µm thickness. The tissue sections were placed onto glass slides and washed three times with distilled water. The air-dried sections were mounted with fluorescence mounting medium and the distribution of the ATTO-labeled liposomes was observed using a confocal laser scanning microscope (FV3000, OLYMPUS, Tokyo, Japan). The measuring sensitivity was set using the untreated left eye of each rat as a negative control. For all samples, the instillation volume into the eye was set 20 µL or 30 µL, which includes appropriate amounts of ATTO-labeled liposomes for the observation of fluorescence in the retinal flat mount or thin eye tissue sections. Evaluation time was set as 2 h, which shows the marked localization of the STR-CH2R4H2C-modified liposomes in a retina (data not shown).

### 2.8. Statistical Analysis

Data are presented as the mean ± standard deviation (SD) (*n* = 3). Statistical analysis was performed using Student’s *t*-test or one-way ANOVA. Statistical significance was defined as *p* < 0.05.

## 3. Results

### 3.1. Physical Properties of the STR-CH2R4H2C-LP-siRNA Lipoplexes

Cationic DOTAP-based liposomes have been widely used as effective nanocarriers for gene delivery. However, these positive-charged carriers often cause cytotoxicity and tissue damage. Thus, we aimed to prepare a hybrid nanocarrier that controlled surface charges and intracellular dynamics using a positively-charged multifunctional peptide (STR-CH2R4H2C) for safe and effective siRNA delivery to the retina. STR-CH2R4H2C (0.0, 1.0, or 2.0 mol% to total lipids) was added to DOPC/CHO-based (neutral-LP) or DOTAP/DOPC/CHO-based (cationic DOTAP-LP; DOTAP, 0.125–0.750 µmol per 3 µmol of total lipid) liposomes. Neutral-LP and cationic-LP have particle sizes around 60–70 nm and different zeta potentials (neutral-LP, −14.4 ± 7.2 mV; DOTAP0.125-LP, +21.9 ± 8.1 mV; DOTAP0.250-LP, +27.0 ± 9.5 mV; DOTAP0.375-LP, +42.9 ± 13.8 mV; DOTAP0.500-LP, +50.7 ± 16.5 mV; DOTAP0.750-LP, +59.2 ± 13.8 mV). Table 2 shows the particle sizes, zeta potentials, and siRNA loading efficiencies of the different STR-CH2R4H2C-LP-siRNA lipoplexes. All of the STR-CH2R4H2C-LP-siRNA lipoplexes had a positive charge. The amount of siRNA loaded in the neutral-LP-based lipoplexes could not be detected due to their weakly negative charge. In the DOTAP0.125-LP-based and DOTAP0.250-LP-based lipoplexes, the siRNA loading efficiencies were higher (around 80–100%) than those in the other cationic-LP-based lipoplexes. Particle sizes of the DOTAP0.125-LP-based lipoplexes increased with an increasing molar ratio of STR-CH2R4H2C, regardless of the positive charge, indicating that an agglomeration formed by adding siRNA. The DOTAP0.250-LP-based 2.0 mol% STR-CH2R4H2C-LP-siRNA lipoplexes had a particle size of 70 to 80 nm without agglomeration and a high siRNA loading efficiency of over 90%.

### 3.2. Cellular Uptake of STR-CH2R4H2C-LP-siRNA Lipoplexes and siRNA Transfection in RPE Cells

The cellular uptake of the STR-CH2R4H2C-LP-siRNA lipoplexes was evaluated in rat RPE cells by flow cytometry. Figure 1 shows the effects of the DOTAP composition ratios in cationic-LP and the STR-CH2R4H2C modification on intracellular uptake and siRNA delivery efficiency. The fluorescence per cell of ATTO-labeled LP and FAM-siRNA are shown in Figure 1a and Figure 1b, respectively. The intracellular uptake of the lipoplexes of DOTAP of 0.375 to 0.750 µmol did not change with differences in the STR-CH2R4H2C modification ratio, whereas the DOTAP0.125-LP-based STR-CH2R4H2C-LP-siRNA lipoplexes and DOTAP0.250-LP-based STR-CH2R4H2C-LP-siRNA lipoplexes demonstrated significantly higher cellular uptake. However, for the DOTAP0.125-LP-based lipoplexes (Figure 1b), there was no increase in siRNA transfection efficiency by the addition of STR-CH2R4H2C even though its uptake increased (Figure 1a). This finding may be because the particle size was large (approx. 108–164 nm) due to electrostatic interactions between the anionic siRNA and cationic DOTAP0.125-LP. In fact, the zeta potentials of DOTAP0.125-LP-based STR-CH2R4H2C-LP-siRNA lipoplexes decreased when loading siRNA compared to the original DOTAP0.125-LP (zeta potential approx. +20 mV, particle size 68.5 ± 30.2 nm). The STR-CH2R4H2C-LP-siRNA lipoplexes consisting of DOTAP0.375-LP, DOTAP0.500-LP, and DOTAP0.750-LP showed no changes in cellular uptake by the STR-CH2R4H2C modification, suggesting that the contribution of the positive charge of DOTAP was superior to that of STR-CH2R4H2C.

The 2.0 mol% STR-CH2R4H2C modification to DOTAP0.250-LP significantly improved the siRNA transfection efficiency (Figure 1b). The STR-CH2R4H2C-LP-siRNA lipoplexes composed of cationic-LPs with DOTAP0.250-LP and DOTAP0.125-LP showed no cytotoxicity by the adenosine triphosphate assay, whereas all lipoplexes composed of DOTAP of 0.375 to 0.750 µmol (per 3 µmol of total lipid in cationic-LP) decreased cell viability by 30% (data not shown). Figure 2 shows the effects of the STR-CH2R4H2C modification and surface charges on the cellular uptake and siRNA transfection efficiency of 2.0 mol% STR-CH2R4H2C-LP-siRNA. In neutral-LP-based lipoplexes, STR-CH2R4H2C effectively improved the intracellular uptake of the lipoplexes by more than 10-fold and significantly improved siRNA transfection efficiency. The STR-CH2R4H2C modification to DOTAP0.250-LP enhanced the intracellular uptake of both lipoplex and siRNA further. Interestingly, the neutral-LP-based 2.0 mol% STR-CH2R4H2C-LP-siRNA lipoplexes showed a similar intracellular uptake to that of the unmodified DOTAP0.250-LP-based lipoplexes; however, the siRNA transfection efficiency was approximately 50% lower. Moreover, the ratio of the siRNA loaded on both the surface of cationic liposomes and the peptide residue of STR-CH2R4H2C was 1 to 1 (approx. 50% each per formulated total siRNA amount), suggesting that the siRNA interacting with STR-CH2R4H2C could perform superior intracellular dynamics. These results demonstrated that a hybrid nanocarrier comprising the STR-CH2R4H2C peptide and cationic liposomes could provide highly efficient siRNA delivery.

### 3.3. Advantages of STR-CH2R4H2C-LP-siRNA Lipoplexes for the Intracellular Dynamics of siRNA and siRNA Activity in RPE Cells

The STR-CH2R4H2C is a cytoplasm-responsive stearylated peptide that promotes high cellular uptake, escape from the endosome by the proton sponge effect, and stable complexes with siRNA that dissociate in the cytoplasm. To exert an effect, siRNA must be released from the endosome into the cytoplasm. As shown in Figure 3, STR-CH2R4H2C-LP-Cy5-siRNA lipoplexes composed of 2.0 mol% STR-CH2R4H2C and DOTAP0.250-LP greatly increase the release of siRNA from the endosomes. The fluorescence of Cy5-siRNA was very low in cells treated with the neutral-LP-based 2.0 mol%STR-CH2R4H2C-LP-siRNA lipoplexes and unmodified cationic DOTAP0.250-LP-sRNA lipoplexes, which may be due to low siRNA transfection.

Furthermore, the intracellular activity of siRNA was evaluated using siVEGF to inhibit the production of VEGF, a factor inducing both neovascularization in retinal disease and nonspecific control siRNA (siControl). RPE cells were incubated with naked siRNA, DOTAP0.250-LP-based 2.0 mol% STR-CH2R4H2C-LP-siRNA lipoplexes, or unmodified DOTAP0.250-LP-siRNA lipoplexes, and VEGF protein and mRNA were quantified by ELISA and RT-qPCR, respectively. The suppression of VEGF protein expression was evaluated using LipoTrust, a commercially available siRNA transfection regent, as a positive control. As shown in Figure 4, STR-CH2R4H2C-LP-siVEGF lipoplexes significantly suppressed VEGF mRNA and protein expression compared to the naked siRNA and unmodified lipoplexes. In addition, 2.0 mol% STR-CH2R4H2C-DOTAP0.250-LP-siRNA lipoplexes showed a VEGF protein knockdown efficacy comparable to that of the LipoTrust, suggesting that the cytoplasm-responsive function of STR-CH2R4H2C in the DOTAP0.250-LP increased transfection efficiency and the intracellular dynamics of siRNA.

### 3.4. Intraocular Migration of STR-CH2R4H2C-LP Following Instillation into Rat Eyes

Tight junction formation in the BRB is a strong barrier to the delivery of drugs and macromolecules to the posterior segment of the eye, particularly the retina, through either systemic circulation or topical administration. Figure 5a shows the images of retinas isolated from rat eyes 2 h after the instillation of neutral STR-CH2R4H2C-LP or cationic STR-CH2R4H2C-DOTAP0.250-LP. The fluorescence of the ATTO647-labeled LP was observed in the 2.0 mol% STR-CH2R4H2C-DOTAP0.250-LP, whereas no fluorescence observed in the tissues with the unmodified DOTAP0.25-LP and neutral STR-CH2R4H2C-LP. Moreover, we performed evaluation of retinal histology after instillation of DOTAP0.250-LP. Figure 5b shows the results to observed using a confocal laser scanning microscope of eye tissue sections removed 2 h after instillation of unmodified DOTAP0.250-LP and 2.0 mol% STR-CH2R4H2C-DOTAP0.250-LP. The fluorescence of the 2.0 mol% STR-CH2R4H2C-DOTAP0.250-LP was found to be present in the posterior segment of the eye, it seems that the peptide-modified liposomes distribute in the outer layer of the retina (i.e., RPE cell layer). These results suggest that the STR-CH2R4H2C-modified cationic liposomes could be used as a nanocarrier for eye drop administration. The single dose instillation into the rat eye was applied in this study because we aimed the preparation and evaluation of functions of the STR-CH2R4H2C-modified liposomes. For clinical use, since frequent instillation is assumed, we are also currently investigating frequent administration to evaluate the suppressing effects using neovascularization model rats.

(a) Effects of STR-CH2R4H2C modification and surface charges on migration into rat retinas following instillation were evaluated. Neutral-LP, neutral 2.0 mol% STR-CH2R4H2C-LP, cationic DOTAP0.250-LP, and cationic 2.0 mol% STR-CH2R4H2C-DOTAP0.250-LP were instilled into the right eyes of rats (Total lipid 0.60 µmol/20 µL/eye). After 2 h, the eyeballs were removed, and retinal flat mounts were prepared. The fluorescence of ATTO-labeled LPs was observed using a fluorescence microscope. Exposure time was set using the untreated left eye of each rat as a control. Magnification: 100×.

(b) ATTO647N-DOPE-labeled DOTAP0.250-LP and 2.0 mol% STR-CH2R4H2C-DOTAP0.250-LP (Lipid, 0.18 µmol/30 µL) were instilled in 30 µL to rat eyes. The eyeballs were removed, and the eye tissue sections were prepared. The liposome distribution of retinal tissue was observed using CLSM. Red color indicates ATTO647N-DOPE. Scale bar: 100 µm, magnification: 200×.

## 4. Discussion

Positively-charged peptides, such as HIV-Tat-derived peptide rich in arginine and R-8 (Arg-Arg-Arg-Arg-Arg-Arg-Arg-Arg) peptide, possess superior cell permeability and gene delivery ability [23,24,25]. We previously synthesized a cytoplasm-responsive stearylated peptide (STR-CH2R4H2C) consisting of arginine, histidine, and cysteine [30]. STR-CH2R4H2C is an effective siRNA carrier that promotes high cellular uptake via arginine, escapes from the endosome by the proton sponge effect of histidine, and forms stable complexes with siRNA that dissociate in the cytoplasm through cysteine [29,32,33]. Liposomes have been widely used as biocompatible drug delivery systems that can carry low-molecular compounds to macromolecules. Their function can be improved by adding various characteristics, such as pH responsiveness and targeting.

Improving intraocular migration and intracellular activity is important for effective siRNA delivery to the posterior segment of the eye after instillation. DOPC/CHO-based cationic liposomes and anionic DOPE/CHEMS liposomes are available carriers that enhance the permeation of siRNA into the skin. Cationic liposomes show higher penetration with an increasing positive surface charge; however, the high positive charge causes skin irritation and cytotoxicity [16]. The purpose of this study was to develop hybrid non-cytotoxicity nanocarriers with effective siRNA intracellular dynamics by controlling surface charges using STR-CH2R4H2C for targeted siRNA delivery to the RPE of the eye.

STR-CH2R4H2C can modify the surface of liposomes through the hydrophobic interactions between the stearyl group of STR-CH2R4H2C and the fatty acid ester residues of the lipid bilayer of the liposome [43,44]. We determined the optimized composition of the hybrid nanocarriers to be 2.0 mol% STR-CH2R4H2C and cationic DOTAP/DOPC/CHO liposomes composed of 0.25 µmol DOTAP (per 3 µmol of total lipid). This nanocarrier could improve the efficacy of siRNA loading and intracellular uptake in retinal cells. Although the surface charge of cationic DOTAP/DOPC/CHO liposomes increased with an increased DOTAP molar ratio, no change in particle size was observed; all particles were 60–70 nm. The STR-CH2R4H2C-modified cationic nanocarrier composed of DOTAP0.125-LP significantly enhanced cellular uptake but did not increase siRNA delivery efficiency due to increased particle size and a low transfected amount of siRNA. In contrast, the STR-CH2R4H2C-modified cationic nanocarriers composed of 0.375 µmol or higher DOTAP (per 3 µmol of total lipid) had high surface charges of more than +40 mV. Contrary to expectation, the efficacy of intracellular transduction of siRNA tended to be reduced by the STR-CH2R4H2C modification, which may be caused by decreased siRNA loading due to electrostatic repulsion between the peptide and liposome.

We evaluated the release of siRNA from the lipoplexes in PBS (pH 7.4) at room temperature, and several percents of loaded siRNA was released from lipoplexes in an hour (data not shown). However, the in vivo–in vitro correlation (IVIVC) on the release of siRNA form lipoplexes in the posterior segment of the eye is unclear, and it is difficult to assess the function of lipoplexes for delivery of siRNA using an in vitro release test. The IVIVC on the releasing profile of the active ingredient is one of issues in the development of nano-DDS products; an establishment of a suitable procedure for in vitro release tests will be necessary for further development.

STR-CH2R4H2C has a high positive charge because it is arginine-rich, which can cause cytotoxicity at high concentrations through its strong interaction with cell surfaces. Although not shown in the data, the STR-CH2R4H2C-modified neutral nanocarrier was cytotoxic at a modification ratio of 3 mol% or higher, with further cell shedding of about 30% at 4 mol%. The composition of 2 mol% peptide-modified nanocarriers, composed of DOTAP0.250-LP and DOTAP0.125-LP, showed no cytotoxicity regardless of whether it was positively charged, whereas the cell viability decreased with increase in the composition ratio of DOTAP. In particular, 2 mol% peptide-modified DOTAP0.250-LP which promoted siRNA efficacy was suitable for safe use. The STR-CH2R4H2C peptide may contribute to reducing the risk of cell and tissue damages by cationic DOTAP liposomes due to its effective functions which show stable complexation with siRNA, no cytotoxicity, and high siRNA transfection efficacy [29,30]. In addition, the 2 mol% peptide-modified DOTAP0.250-LP observed showed no morphological changes in the eye tissue sections in its instillation (data not shown), demonstrating the application is safe for the living body when used in an appropriate concentration. Several reports have indicated that neutral liposomes must be modified with ligands or cationic lipids to enhance their interactions with the cell surface [45,46,47]. The DOPC/CHO liposomes used in this study had a neutral or weakly negative charge. The surface charges of the neutral liposomes were slightly increased with an increase in the modification ratio of STR-CH2R4H2C, but all of which were negatively charged. Despite the low surface charge, the STR-CH2R4H2C-modified DOPC/CHO nanocarrier demonstrated significant cellular uptake, suggesting that STR-CH2R4H2C enhances the interaction between the liposomes and the plasma membrane, regardless of the surface charge [37]. Furthermore, the STR-CH2R4H2C-modified DOPC/CHO nanocarrier had a lower surface charge (about +20 mV) than DOTAP0.25/DOPC/CHO-LP, but its intracellular uptake efficiency was comparably high. However, its siRNA transfection efficiency was lower than DOTAP0.25/DOPC/CHO-LP. These differences may be because the STR-CH2R4H2C-modified DOPC/CHO nanocarrier is negatively charged, resulting in lower siRNA loading efficiency than DOTAP0.25/DOPC/CHO-LP. In the DOTAP0.25/DOPC/CHO-based nanocarriers, both STR-CH2R4H2C-modified and unmodified carriers had positive surface charges, but the cellular uptake efficiencies differed by more than 2-fold. The quaternary ammonium group of DOTAP contributes to the internalization of the liposomes by acting on peptidoglycans and other components of the plasma membrane [48]. The guanidino group of arginine in STR-CH2R4H2C not only interacts strongly among liposomes, cell membranes, and siRNA, but the other amino acid residues of STR-CH2R4H2C also play a role in effective siRNA delivery [49,50].

Nanocarriers are taken up into cells by endocytosis. During endocytosis, a part of the cell membrane deforms and collapses in a way that envelops a substance from the outside, internalizing the substance in the endosomal membrane. Endosomes are weakly acidic (pH 5.5) and rich in enzymes involved in the metabolism and elimination of substances [51,52]. Therefore, escaping from the endosome is essential for siRNA activity. The STR-CH2R4H2C peptide enhances the escape of siRNA from the endosomes (Figure 3). The basic amino acids arginine and histidine alter the pH environment within the endosomes by protonation under acidic conditions. An influx of protons into the endosome causes the endosomal membrane to explode due to an increase in osmotic pressure, releasing siRNAs into the cytoplasm [53,54]. We demonstrated that the 2.0 mol% STR-CH2R4H2C-modified DOTAP0.25/DOPC/CHO liposomes were the optimal nanocarrier, combining high siRNA loading, high cellular uptake of siRNA, and enhanced siRNA release from endosomes. For this nanocarrier, it was estimated that approximately equal amounts of siRNA were loaded on both the liposome surface and the peptide. We speculate that the siRNA loaded on the peptide could contribute to the improved siRNA intracellular dynamics. In fact, the nanocarrier-loaded siVEGF showed high suppressive efficacies against VEGF mRNA and protein expression (Figure 4). The unmodified DOTAP0.25/DOPC/CHO liposomes had no significant suppressive efficacies, which may be because siRNA transfection and endosomal escape were low in this nanocarrier.

The blood–brain barrier and BRB are the main cause of poor drug delivery to the respective organs. Thus, improving the migration of drug molecules in the target tissue is an important issue. Retinal diseases, such as age-related macular degeneration, are the leading cause of blindness and low vision. Due to the BRB, drug therapy for retinal diseases is mainly administered by invasive intravitreal injection. Macromolecules, such as siRNA, are particularly difficult to deliver to the retina by intraocular delivery via eye drops. We previously reported that transferrin-modified liposomes of 80 nm or less allowed permeation through the fenestrated structure of the choroidal capillary and improved the delivery efficiency to the posterior segment of the eye using eye drops [55]. The 2.0 mol% STR-CH2R4H2C-modified DOTAP0.25/DOPC/CHO liposome has a small particle size of approximately 70 nm even after complexation with siRNA. As shown in Figure 5a, the STR-CH2R4H2C-modified DOTAP0.25/DOPC/CHO carrier could migrate into the retina after administration through non-invasive eye drops. In contrast, the STR-CH2R4H2C-modified DOPC/CHO nanocarrier and unmodified DOTAP0.25/DOPC/CHO liposomes showed no migration into the retina after topical treatment. Additionally, the fluorescence of 2.0 mol% STR-CH2R4H2C-DOTAP0.250-LP was observed in the outer layers of the retina (Figure 5b). There are RPE cells in this layer that are responsible for BRB function, and it may be possible to improve the BRB permeability of drugs using these peptide-modified liposomes. Cationic liposomes are known to enhance retention in the eye due to their interactions with negatively charged factors, such as mucopolysaccharides and glycosaminoglycans in the sclera [56,57]. The cornea has a tighter structure and lower molecule permeability than the sclera, which has a relatively thin segment (thickness; 1.0–1.3 mm around the optic nerve and 0.3–0.5 mm around the corneal junction and equator) [57,58]. Permeability in the sclera is affected by molecular weight, molecular radius, and hydrophilicity [59,60]. Although the detailed delivery route of STR-CH2R4H2C-modified DOTAP0.25/DOPC/CHO nanocarrier is unclear, we hypothesize that this nanocarrier penetrates the retina through routes other than the cornea after instillation. Various CPPs have been widely used to improve intracellular transduction of low- and high-molecular-weight compounds. The STR-CH2R4H2C peptide can enhance the functions of liposomes for siRNA delivery. Currently, Antibody drugs have been used clinically for retinal diseases. However, it is an invasive drug therapy that is injected into the vitreous, which has severe risks (e.g., retinal detachment, infection, and physical and mental burden on the patient). We have focused on nucleic acid medicines (e.g., siRNA, antisense RNA) which are expected to be the next generation following antibody medicines, capable of chemical synthesis and modifications, and adding various functions. We have aimed to suppress VEGF with siRNA, but also expect to apply other kinds of siRNAs and combination therapy with siRNA inhibiting bioactive factors (e.g., inflammatory cytokines) related to disorders in retina. The hybrid nanocarrier based on these multifunctional peptides and cationic liposomes could be a useful carrier for non-invasive siRNA therapy for intractable ocular diseases.

## 5. Conclusions

The multifunctional STR-CH2R4H2C peptide improves the efficiency of the cellular uptake of nanocarriers, intracellular dynamics of siRNA, and intraocular migration after eye drop administration. A hybrid nanocarrier consisting of a multifunctional peptide and cationic liposomes represents a promising approach for non-invasive intraocular siRNA delivery to the retina.

## Data Availability

Not applicable.

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
