# Peer review of "A Multifunctional Hybrid Nanocarrier for Non-Invasive siRNA Delivery to the Retina"

_pharmaceutics, 2023, doi:10.3390/pharmaceutics15020611_

Round 1

Reviewer 1 Report

The authors developed a STR-CH2R4H2C-modified cationic nanocarrier aiming to deliver siRNA into retina via eye drops. Why the aim is interesting and there are some data supporting 2 mol% STR-CH2R4H2C-modified cationic nanocarrier slightly increased siRNA delivery efficiency to cultured RPE cells, they did not provide convincing data showing that this method could effectively deliver siRNA into retina. Please address the following comments:

1. The efficiency of siRNA transfection and knockdown of a gene expression was low (Fig. 3 &4). Please include several commercially available siRNA transfection methods (e.g. Lipofectamine 3000, Lipofectamine RNAiMAX, hiperfect transfection reagent, etc) in the experiments to compare if CH2R4H2C-modified cationic nanocarrier was doing better job to delivery siRNA into cells than these reagents or not.

2. In Fig.5, there were only three red dots in cationic 2.0 mol% STR-CH2R4H2C-DOTAP0.250 LP-siRNA treated eyes. The dots could just be some non-specific fluorescent particles. Even if they were siRNA infected cells, what cell types were they? It would be help to stain retinal flatmount and retinal sections with cell markers to determine cells transfected with siRNA. Nonetheless, since the transfection efficiency was so low, it would be useless for treating any retinal disease.

3. In 2.1. Reagents and cells. Please list catalogue number for each reagent used to generate nanocarriers.

Author Response

We sincerely appreciate you spending your time and effort to review this manuscript. Thank you very much for giving us useful comments. We have carefully reviewed and answered the concerns of each reviewer.

Reviewer 2 Report

It is interesting to study non-invasive instillation in ocular drug research. In this study, the authors used STR-CH2R4H2C peptide to delivery VEGF siRNA in RPE cells. The authors show the delivery efficiency in cells/retina tissue, and VEGF knockout efficiency in cells. However, there are several limitations.

1, I would suggest the authors to testify the safety of STR-CH2R4H2C and STR-CH2R4H2C-LP in vivo and in vitro.

2, The production of VEGF is dynamic. The clinical drug target is VEGF receptor. Please discuss the prospects for VEGF siRNA in clinical application.

3, Is there any VEGF-relative animal model could be used in this study? Such as diabetic retinopathy and AMD models.

4, The authors used 20 μL STR-CH2R4H2C-LP (ATTO-DOPE 30 μM) as eye drops. Why use 20 μL on rats? Why examine the delivery efficiency after 2 hours? Please explain the method issue.

5, In figure 5, the authors examine the delivery efficiency on retinas, however, they testify the delivery efficiency in RPE cells in figure 2-4. It is not matched from in vivo to in vitro studies.

6, In table 2, please explain why siRNA loaded efficiency could over 100%.

7, Line 351, figure 4 should be figure 5.  

Author Response

(The authors gave the same response as above.)

Reviewer 3 Report

In this research article, the authors presented “A multifunctional hybrid nanocarrier for non-invasive siRNA delivery to the retina”. From my point of view, the topic is fascinating. The manuscript is concise and well-written. However, the manuscript has some issues which need to be addressed before its publication in Pharmaceutics. Following are my concerns:

1)     Authors should write the product number and the proper amount of all the chemicals used in this study so that others can repeat the synthesis.

2)     Has someone used STR-CH2R4H2C for siRNA delivery to the retina?

3)     What will happen if STR-CH2R4H2C concentration increases by more than 2%?

4)     The fluorescence in figure 4 seems very low. Is there any positive control by which one can compare the delivery to the retina?

Author Response

(The authors gave the same response as above.)

Reviewer 4 Report

The submitted manuscript reports on the development of a nanocarrier for the non-invasive delivery of small interfering RNA (siRNA) to the posterior segment of the retina by eyedrops. This topic is of interest for readers of Pharmaceutics. However, I have some reservations about of the experimental design and data presented. I therefore recommend publication of this manuscript only if the authors can address the major issues noted below.

1. The authors characterized the lipoplexes with dynamic light scattering. It is better to confirm the morphology of the lipoplexes using a transmission electron microscope as well.

2. The added peptide was mentioned in the manuscript. Have the peptide amounts in different lipoplexes been quantified to confirm their theoretical amounts? Please provide relative data.  

3. Although siRNA loading efficiency is provided, release profiles of the formulation need to be analyzed as well. I suggest the authors carry out a release kinetic study.

4. Cellular uptake study using flow cytometry showed the uptake of the siRNA in the three formulations. However, in the confocal results, it seems only 2.0 mol% STR-CH2R4H2C-DOTAP0.250-LP-333 siRNA lipoplexes distributed in the cells. Need to check the data or explain the reason. Also one group (0 mol% STR-CH2R4H2C-neutral-LP) is missing.

5. For cellular studies, can the authors reproduce the graphs by normalizing each value on the cell number? Probably it could be easier to understand.

6. In Figure 4, control groups (blank and naked siRNA) are not studied. Proper control groups are needed to compare the results.

7. Histological analysis of tissue samples may be required.

8. The information ins some figures is difficult to read and need to increase the size and resolution.

Author Response

(The authors gave the same response as above.)

Round 2

Reviewer 1 Report

The authors have addressed my concerns.

Reviewer 4 Report

The authors have addressed my comments and I am happy to endorse it for publication.